# Single-molecule studies contrast ordered DNA replication with stochastic translesion synthesis

**Gengjing Zhao, Emma S Gleave, Meindert Hugo Lamers†***

MRC laboratory of Molecular Biology, Cambridge, United Kingdom

**Abstract** High fidelity replicative DNA polymerases are unable to synthesize past DNA adducts that result from diverse chemicals, reactive oxygen species or UV light. To bypass these replication blocks, cells utilize specialized translesion DNA polymerases that are intrinsically error prone and associated with mutagenesis, drug resistance, and cancer. How untimely access of translesion polymerases to DNA is prevented is poorly understood. Here we use co-localization single-molecule spectroscopy (CoSMoS) to follow the exchange of the *E. coli* replicative DNA polymerase Pol IIIcore with the translesion polymerases Pol II and Pol IV. We find that in contrast to the toolbelt model, the replicative and translesion polymerases do not form a stable complex on one clamp but alternate their binding. Furthermore, while the loading of clamp and Pol IIIcore is highly organized, the exchange with the translesion polymerases is stochastic and is not determined by lesion-recognition but instead a concentration-dependent competition between the polymerases.
DOI: https://doi.org/10.7554/eLife.32177.001

**\*For correspondence:**
m.h.lamers@lumc.nl

**Present address:** †Leiden University Medical Center, Leiden, Netherlands

**Competing interests:** The authors declare that no competing interests exist.

## Introduction

To ensure faithful replication of the genomic DNA, replicative DNA polymerases have a narrow active site that limits the incorporation of incorrect nucleotides. In addition, rare nucleotide misincorporations into the primer strand prevent further DNA synthesis and a 3′−5′ exonuclease is required to remove the misincorporated nucleotides (*Kunkel, 2004*). In contrast, when the polymerase encounters a lesion on the template strand in the form of a modified base caused by diverse chemicals, reactive oxygen species, or UV light (*Liu et al., 2016*; *Lindahl, 1996*), the high-fidelity replicative DNA polymerases are stalled. To bypass these replication blocks, all cells harbor multiple specialized translesion DNA polymerases (*Goodman and Woodgate, 2013*) that have more open active sites and are therefore able to accommodate bulky DNA adducts and continue DNA synthesis. As a result of their more open active sites, the translesion polymerases are error prone and consequently associated with increased mutagenesis, drug resistance, and cancer (*Fuchs and Fujii, 2013*; *Lange et al., 2011*). Therefore, the access of the translesion polymerases to DNA needs to be tightly controlled, but how this is achieved has been the subject of debate.

The 'toolbelt' model (*Indiani et al., 2005*) predicts that in *E. coli* the replicative DNA polymerase Pol IIIα and the translesion DNA polymerase Pol IV bind simultaneously to the DNA sliding clamp β, a dimeric, ring-shaped protein that encircles the DNA and provides processivity to the replicative DNA polymerases (*Johnson and O'Donnell, 2005*). This way, the translesion polymerase functions in a manner analogous to the proofreading exonuclease: when the replicative polymerase inserts an incorrect nucleotide into the nascent strand, it will be removed by the proofreading exonuclease, whereas when the polymerase encounters a lesion on the template strand, the DNA is transferred to the translesion polymerase that can bypass the lesion. Thus both the exonuclease and translesion polymerase act as 'tools' that enable the replicative polymerase to overcome potential roadblocks to DNA replication. The toolbelt model, which was originally based on steady-state Förster Energy

Resonance Transfer (FRET) experiments that showed the simultaneous binding of the replicative and translesion polymerases to the β-clamp, has found support in several subsequent studies (*Indiani et al., 2009*; *Furukohri et al., 2008*; *Heltzel et al., 2012*). However, all these studies used bulk studies that due to the asynchronous nature cannot separate out the sequential steps during a reaction (*Robinson and van Oijen, 2013*). More recently, single molecule approaches have also been used (*Kath et al., 2014*, *2016*), but in these experiments the exchange of polymerases was inferred indirectly through the change in speed of DNA synthesis and therefore it cannot be determined whether the polymerases bind simultaneously. In addition, recent studies reveal that during DNA replication in *E. coli* the two binding pockets of the dimeric β-clamp are occupied by the replicative DNA polymerase Pol IIIα and the associated proofreading exonuclease ε (*Toste Rêgo et al., 2013*; *Jergic et al., 2013*). The cryo-EM structure of the trimeric Pol III-exonuclease-clamp (α, ε, β) complex (*Fernandez-Leiro et al., 2015*) also shows that most of the clamp is covered, leaving no space for a second polymerase. Therefore, it remains controversial whether the replicative and translesion polymerases can co-localize on a single clamp.

Consequently, an alternative view to the toolbelt model is that the translesion DNA polymerases compete for binding to clamp-DNA through 'mass action', as evidenced by the fact that the bypass of a $N^2$-acetylaminofluorene guanine adduct by Pol V or Pol II depends on the relative concentrations of the two polymerases (*Becherel and Fuchs, 2001*; *Fujii and Fuchs, 2004*), and that the concentrations of Pol IV and Pol V are dramatically increased during the bacterial SOS DNA damage response (*Sutton, 2010*).

Regardless of the model, the DNA sliding clamp plays a pivotal role in controlling access of the translesion polymerases to the DNA. However, the control for access to the clamp-DNA is complicated by the fact that on the lagging strand, DNA synthesis is discontinuous and every ~1000 base pairs a new clamp is loaded, followed by the binding of the replicative DNA polymerase. Due to its closed circular shape, the clamp must be loaded and unloaded onto the DNA by the dedicated clamp loader complex (γ/τ-complex in bacteria, RFC in archaea and eukaryotes) (*Hedglin et al., 2013*; *Kelch et al., 2012*). Once the clamp is loaded onto primed DNA, the replicative polymerase associates and initiates DNA synthesis. How the repeated loading and unloading of the clamp and replicative polymerase Pol IIIα on the lagging strand is coordinated, while simultaneously preventing the untimely association of the translesion polymerases has not been studied so far.

Here, we use co-localization single molecule spectroscopy (CoSMoS) (*Friedman et al., 2006*) to directly visualize the loading of the *E. coli* clamp loader (γ/τ-complex), the DNA sliding clamp β, the replicative DNA polymerase Pol IIIα, the proofreading exonuclease ε, as well as the exchange with the translesion polymerases Pol II and Pol IV. The multi-color CoSMoS experiments enable us to follow the binding and dissociation of multiple proteins in real-time on a single DNA molecule, which makes it the most suitable method to discriminate between simultaneous or sequential binding of different molecules on a DNA substrate. Our work shows that the translesion polymerases Pol II and Pol IV do not form a stable complex with the replicative polymerase Pol IIIα on the clamp-DNA and therefore the clamp does not function as a molecular toolbelt. Furthermore, we find that the sequential activities of the replication proteins clamp loader, clamp, and Pol IIIα are highly organized while the exchange with the translesion polymerases Pol II and Pol IV is disordered and determined by mass action through concentration-dependent competition for the hydrophobic groove on the surface of the β-clamp. Hence, our results provide a unique insight into the temporal organization of the events in DNA replication and translesion synthesis, and contrast the highly organized replication events with stochastic polymerase exchange during translesion synthesis.

## Results

### Preparation of DNA substrates and fluorescently labeled proteins

The ring-shaped *E. coli* β-clamp is capable of threading and unthreading itself on free DNA ends and therefore we attached a primer-template DNA substrate to a glass surface and blocked its free end with monovalent streptavidin (*Figure 1A*). Subsequently, the binding of fluorescently labeled proteins to DNA was followed by two- or three-color total internal reflection fluorescence microscopy (see Materials and methods) with a frame rate of 0.44 and 0.66 s (s), respectively, on ~800 well separated DNA molecules per field of view (*Figure 1B–C*). Proteins were fluorescently labeled via

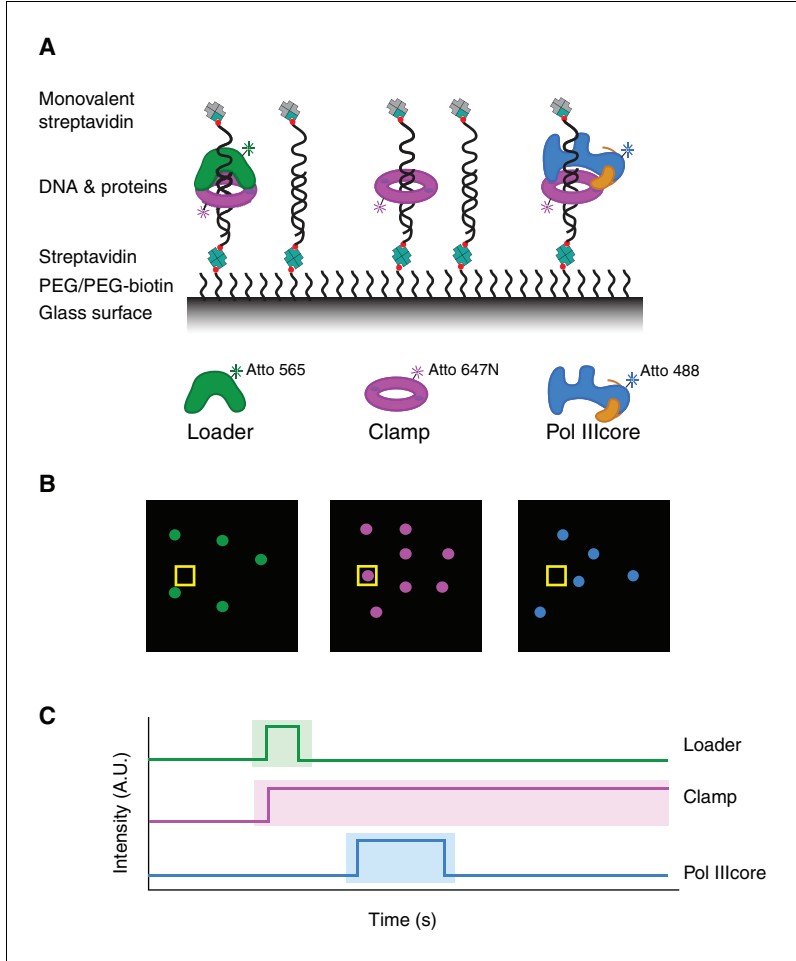

**Figure 1.** Experimental setup. (**A**) Schematic representation of the experimental setup. DNA molecules are attached to a PEGylated glass slide via a biotin-streptavidin layer, and end-blocked with monovalent streptavidin. Fluorescently labeled proteins will be detected when bound to the DNA molecules (**B**) Schematic representation of the three fluorescent channels from a single image (out of an 1000 image series) showing the presence of different molecules. (**C**) Schematic representation of kymographs from single position in the image series, revealing the binding and release of different proteins to the DNA at that position.

DOI: https://doi.org/10.7554/eLife.32177.002

The following figure supplement is available for figure 1:

**Figure supplement 1.** Validation of proteins, fluorophores and DNA.

DOI: https://doi.org/10.7554/eLife.32177.003

maleimide-cysteine crosslinking or enzymatically via an N-terminal Ybbr tag (*Yin et al., 2006*), and the fluorescently labeled proteins retained wild-type activity as indicated by polymerase processivity assay (*Figure 1—figure supplement 1A–C*). For detection of Pol IIIcore ($\alpha$, $\varepsilon$, $\theta$), we fluorescently labeled the $\alpha$ subunit. For the clamp loader complexes ($\gamma_3\delta\delta'$ and $\tau_3\delta\delta'$), the fluorescent label was placed on the $\delta'$ subunit. The lifetime of the each of the fluorophores was measured individually on DNA-bound clamps (Atto 488 274.4 ± 16.5 s, Atto 565: 145.7 ± 7.5 s, Atto 647N: 93.0 ± 7.4 s) (*Figure 1—figure supplement 1D–F*). In most experiments, after initial detection of the DNA molecules, the fluorophore (Atto 488) on the DNA was bleached so that the same color could be re-used on one of the proteins. The bleaching of the DNA fluorophore has no effect on the lifetime of Pol IIIcore on clamp-DNA ($\tau_{on}$ without bleaching 18.1 ± 1.6, $\tau_{on}$ with bleaching 16.8 ± 1.8) (*Figure 1—figure supplement 1G–H*).

## Clamp loading and unloading are distinct processes

The isolated β-clamp shows no interaction with the end-blocked DNA (*Figure 2—figure supplement 1A–B*) When combined with the γ clamp loader complex ($\gamma_3\delta_1\delta'_1$), frequent clamp loading events are observed where the loader and clamp arrive at the DNA simultaneously or in two adjacent frames (*Figure 2A–B*), due to the sequential data acquisition of the three laser channels (further explained in *Figure 2—figure supplement 1C–E*). Therefore, the loader and clamp bind DNA as a pre-formed complex. Shortly after DNA binding, the loader dissociates while the clamp remains bound to the DNA for the remainder of the data acquisition (*Figure 2A*). As the two lifetimes of the loader and clamp are vastly different it was not possible to accurately measure both in one experiment. Therefore, for the clamp lifetime, we first loaded the clamp, washed away the loader and then started data collection using 10 s intervals between measurements to avoid bleaching of the fluorophore. The clamps remain bound for more than 23 min ($\tau_{on}$ = 1429.7 ± 177.0 s; *Figure 2C*), excluding the time it takes to load the clamps, wash away the loader, and start data collection (~3 min). In contrast, the lifetime of the loader is very short lived. To accurately measure the loader lifetime, only one fluorescent channel was used to decrease the frame rate to 0.086 s. The lifetime of the isolated loader on DNA is 1.20 ± 0.05 s, which is shortened to 0.41 ± 0.01 s in the presence of the clamp (*Figure 2D*). The rapid release of the loader from clamp-DNA is dependent on ATP hydrolysis as evidenced by the fact that in the presence of the poorly hydrolysable analog ATPγS, the loader and clamp still bind to the DNA together but also release together (*Figure 2E*). The loader and clamp bind to the DNA briefly ($\tau_{on}$ = 2.7 ± 0.2 s; *Figure 2F*), which contrasts with the long lifetimes for the loaded clamps on DNA (*Figure 2C*). Taken together, our analysis is in agreement with the model that the clamp holds the loader in a conformation that suppresses ATPase activity and subsequent DNA binding triggers ATP hydrolysis and the release of the loader from clamp and DNA (*Hedglin et al., 2013*; *Kelch et al., 2012*).

During DNA replication, the clamps on the lagging strand need to be unloaded and recycled to allow for continuous DNA synthesis (*Yao et al., 1996*). We observe a different temporal organization for unloading events compared to the loading events: the loader arrives at the loaded clamp, releases the clamp within 4.1 ± 0.4 s of its arrival but remains bound for a total time of 10.8 ± 1.2 s (*Figure 2G–I*). Furthermore, unlike clamp loading, the unloading of a clamp does not require ATP hydrolysis: the nucleotide requirement of the unloading process was tested by first loading clamps onto DNA and then washing away the ATP and free proteins. Next, the loader was re-introduced in the absence of ATP, resulting in many unloading events with similar kinetics to those observed in the presence of ATP (*Figure 2J–L*). About half (53%) of the binding events of the loader to the pre-loaded clamp do not result in unloading of the clamp. These events last shorter ($\tau_{on}$ = 2.5 ± 0.1 s) than the unloading events ($\tau_{on}$ = 10.8 ± 1.2 s) (*Figure 2—figure supplement 1F–G*). Taken together, the data indicates that the loading and unloading of a clamp are not forward and backward reactions of the same mechanism but that they are separate processes, each with a distinct organization, possibly to prevent unwanted clamp unloading at the replication fork.

## Pol IIIcore has an intrinsic lifetime on DNA that is independent of its activity

Once the clamp is loaded onto DNA, the replicative DNA polymerase Pol IIIcore will associate with the clamp and initiate DNA replication (*Johnson and O'Donnell, 2005*). Pol IIIcore is a stable trimeric complex containing the polymerase subunit α, the exonuclease ε and the accessory subunit θ (*McHenry and Crow, 1979*). Using a single fluorescent channel, we find that Pol IIIcore alone binds DNA very briefly for one image frame (0.086 s) or less, resembling more a collisions-like interaction rather than true binding events (*Figure 3A–B*). These short-lived collisions are in agreement with previous studies that show that Pol IIIα is a poor enzyme in isolation (*Fay et al., 1981*) that has a low affinity for DNA (*Fernandez-Leiro et al., 2015*; *McCauley et al., 2008*). The collisions of Pol IIIcore contrasts with the well-studied *E. coli* DNA polymerase I Klenow fragment, which is known to have a high affinity for DNA (*Kuchta et al., 1987*) and consequently shows a lifetime of 42.2 ± 1.8 s (*Figure 3C–D*).

The behavior of Pol IIIcore is dramatically altered in the presence of the clamp. Shortly after the clamp is loaded onto DNA, Pol IIIcore associates with the clamp-DNA, producing long lasting binding events (*Figure 3E–G*) with a lifetime of 15.7 ± 1.1 s (*Figure 3E–F*). During DNA replication, the

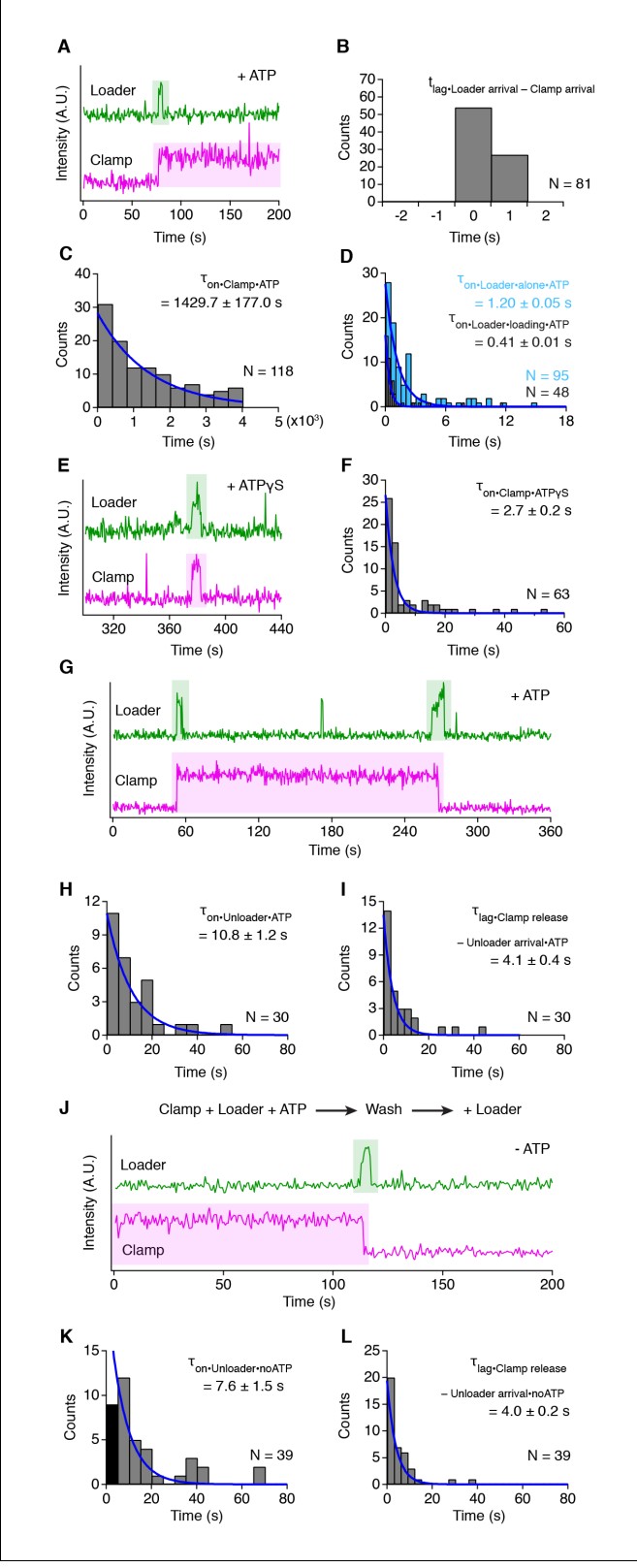

**Figure 2.** Different mechanisms for clamp loading and unloading. (**A**) Representative trace showing a clamp loading event on DNA in the presence of ATP. (**B**) Histogram showing the simultaneous arrival of loader and clamp on DNA (see also *Figure 2—figure supplement 1C–E*). (**C**) The distribution of lifetimes for the clamp on DNA after removal of the clamp loader (**D**) Lifetime of the clamp loader on DNA in the absence (blue bars) and

*Figure 2 continued on next page*

*Figure 2 continued*

presence (grey bars) of the clamp. (**E**) Representative trace showing the simultaneous arrival and release of loader and clamp on DNA in the presence of ATPγS. (**F**) The distribution of lifetimes for the loader and clamp on DNA in the presence of ATPγS. (**G**) Representative trace showing clamp loading and unloading by the loader in the presence of ATP. (**H**) The distribution of lifetimes for the loader on DNA during unloading. (**I**) The distribution of lag times between the arrival of the loader and the release of the clamp. (**J**) Representative trace showing unloading of a pre-loaded clamp in absence of ATP. (**K**) The distribution of lifetimes for the loader during clamp unloading in the absence of ATP. The first column (in dark grey) has been excluded from the fitting. The lower numbers in this column are possibly caused by the clamp that needs to be removed first before the loader can release. (**L**) The distribution of lag times between the arrival of the loader and the release of the clamp in the absence of ATP. All values represent mean lifetime/lag time ±s.e.m.

DOI: https://doi.org/10.7554/eLife.32177.004

The following figure supplement is available for figure 2:

**Figure supplement 1.** Clamp loading and clamp unloading.

DOI: https://doi.org/10.7554/eLife.32177.005

---

*E. coli* replisome synthesizes DNA with up to 100,000 base pairs per binding event (*Yao et al., 2009*; *Tanner et al., 2009*), which contrasts with the relatively short binding times of Pol IIIcore we measure on clamp-DNA. Therefore, we tested the effect of nucleotides on the lifetime of Pol IIIcore on DNA, using only two of the four nucleotides to prevent the Pol IIIcore from extending the primer strand and dissociating at the end the DNA substrate. The omission of two nucleotides is often used in DNA replication assays to induce 'polymerase idling' at a primer terminus as a result of opposing polymerase and exonuclease activities. To our surprise however, addition of dATP and dTTP (0.5 mM each), has no significant effect on the lifetime of Pol IIIcore on clamp-DNA ($\tau_{on}$ = 16.1 ± 1.0 s, *Figure 3G*). This is also in agreement with the observation that an actively synthesizing Pol IIIcore has a lifetime of ~10 s in the presence of all four nucleotides on a 48.5 kb λ DNA substrate (*Jergic et al., 2013*; *Tanner et al., 2008*). Together, this shows that Pol IIIcore has a similar lifetime on clamp-DNA regardless of whether it is stationary, idling or actively synthesizing DNA.

During DNA replication, Pol IIIcore is tethered to the rest of the replisome via the clamp loader protein τ (*Studwell-Vaughan and O'Donnell, 1991*). Thus far, to separate clamp loading from polymerase loading, we have used a clamp loader complex comprising the subunits γ, δ, an δ' ($\gamma_3\delta_1\delta'_1$). γ is a shorter product of τ that is fully active in clamp loading, but does not interact with the polymerase (*Dallmann et al., 1995*, *2000*). To measure the effect of full length τ on clamp loading and polymerase loading, we also measured clamp and polymerase loading with the τ clamp loader complex ($\tau_3\delta_1\delta'_1$), in which each of the three τ proteins had a Pol IIIcore complex bound (see Material and Methods). Here, we find that clamp, loader, and Pol IIIcore arrive at the DNA together (*Figure 3H*). Unlike the γ clamp loader complex, the τ clamp loader complex does not dissociate immediately, as it remains bound for ~15 s and now leaves together with Pol IIIcore ($\tau_{on}$ = 14.8 ± 0.9 s, *Figure 3H–I*). We believe this is a result of the τ clamp loader complex and Pol IIIcore trading places on the clamp-DNA (*Figure 3J*). The presence of the τ, however, does not affect the lifetime of Pol IIIcore on clamp-DNA, as the lifetime of the τ clamp loader - Pol IIIcore complex is similar to that of Pol IIIcore alone.

However, during the simultaneous synthesis of leading and lagging strand, the polymerases on either strand are linked together by the multiple τ proteins of the clamp loader complex, which prevents the polymerase from diffusing away and enables it to quickly resume DNA synthesis. This may explain the higher processivity measured for the replisome (*Yao et al., 2009*; *Tanner et al., 2009*) compared to the bursts of activity from a single Pol IIIcore (*Jergic et al., 2013*; *Tanner et al., 2008*).

## Pol IIIcore and Pol IV alternate binding to clamp-DNA

During DNA replication the replisome may encounter DNA adducts caused by diverse chemicals, reactive oxygen species or UV-light that form barriers to the high-fidelity replicative DNA polymerases. To overcome these replication blocks, the replicative DNA polymerases temporarily trade places with the error-prone translesion DNA polymerases (*Goodman and Woodgate, 2013*). To study the molecular mechanism of polymerase switching, we directly visualized the binding of Pol IIIcore and the translesion DNA polymerase Pol IV on clamp-DNA. At equal concentration of Pol IIIcore and

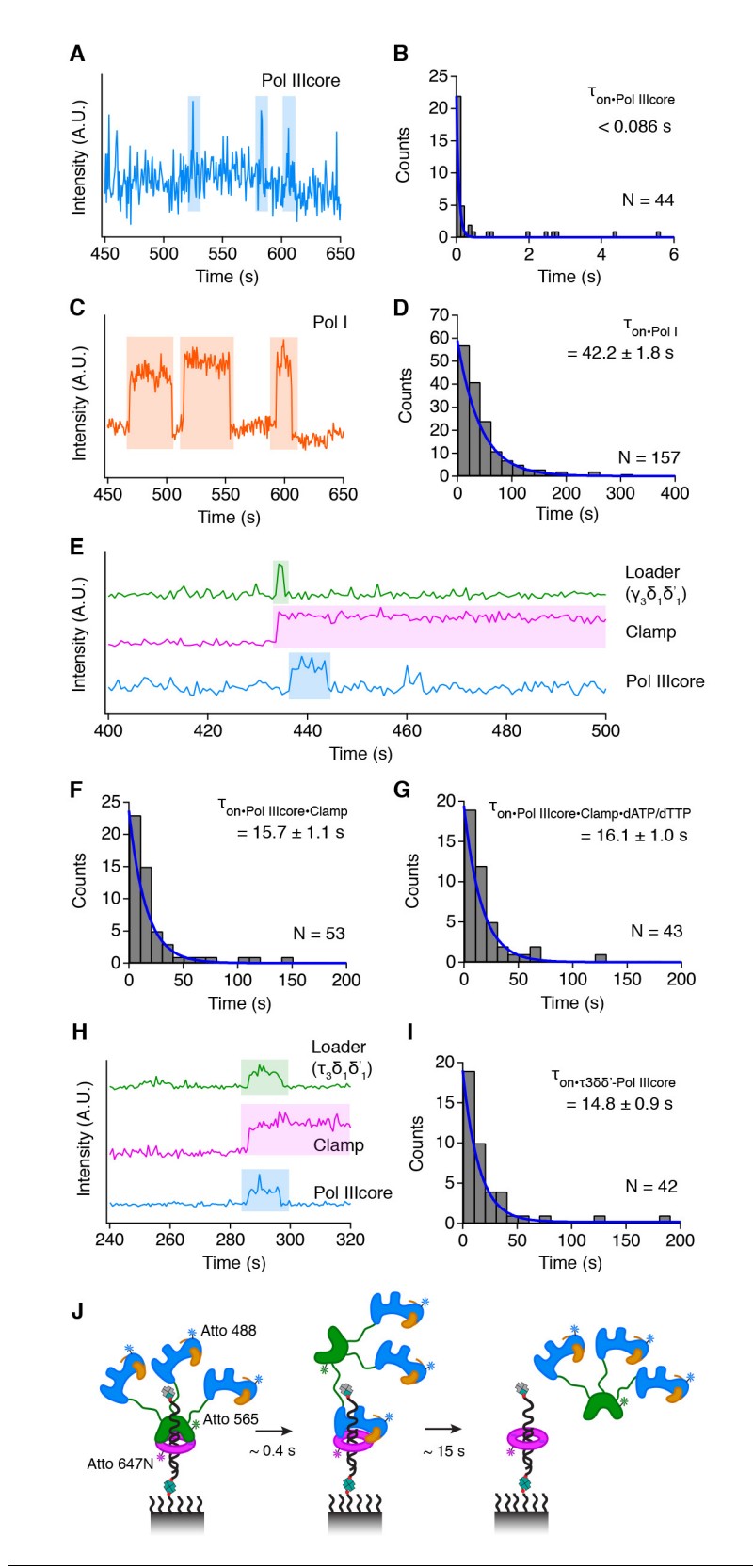

**Figure 3.** Pol IIIcore binds transiently to clamp-DNA. (**A**) Representative trace showing Pol IIIcore collisions with DNA in the absence of clamp. (**B**) The distribution of lifetimes for Pol IIIcore on DNA. (**C**) Representative trace
*Figure 3 continued on next page*

*Figure 3 continued*
showing Pol I binding on DNA. (D) The distribution of lifetimes for Pol I on DNA. (E) Representative trace showing Pol IIIcore binding to clamp-DNA shortly after the release of loader. (F) The distribution of lifetimes for Pol IIIcore on clamp-DNA in absence of dNTPs. (G) The distribution of lifetimes for Pol IIIcore binding events on clamp-DNA in the presence of dATP/dTTP. (H) Loading of clamp and polymerase by the τ clamp loader complex. (I) Lifetime of the τ clamp loader complex and Pol IIIcore on clamp-DNA (J) Cartoon of the binding sequence of the τ clamp loader - Pol IIIcore complex on clamp-DNA. All values represent mean lifetime ±s.e.m.
DOI: https://doi.org/10.7554/eLife.32177.006

Pol IV (30 nM each), we record a large majority of events (92%) that show alternating binding of the two polymerases on clamp-DNA, with Pol IIIcore binding first in 70%, and Pol IV binding first in 22%, of events (*Figure 4A and C*, *Table 1*). Similar to the lifetime of Pol IIIcore alone on clamp-DNA, this competition is unaffected by addition of nucleotides (dATP/dTTP, 0.5 mM each) (Pol IIIcore to Pol IV switch: 72%, Pol IV to Pol IIIcore switch 21%). In the polymerase switching events, there is a significant lag time between the release of Pol IIIcore and Pol IV arrival that decreases with increased protein concentrations (*Table 1*, *Figure 4—figure supplement 1A–D*). The lifetime of Pol IIIcore on the clamp-DNA remains unchanged at all protein concentrations (*Table 1*, *Figure 4—figure supplement 1E–H*), suggesting that Pol IV binding does not cause the release of Pol IIIcore and therefore the two polymerases bind independently.

In addition to the switching events, we also observe a small number of co-localization events of the two polymerases on clamp-DNA (9%), which become more frequent at higher protein concentrations (*Figure 4C*, *Table 1*) but are unaffected by addition of nucleotides (7% with dATP/dTTP). In all events where Pol IIIcore and Pol IV co-localize on the clamp-DNA, the polymerases arrive and leave independently of one another (*Figure 4D*). This is different from the true binding partner exonuclease ε that arrives and leaves with Pol IIIα (*Figure 4F*). In addition, the co-localization time of Pol IIIcore and Pol IV ($\tau_{colocalize}$ = 8.2 ± 0.6, *Figure 4E*) is shorter than that of Pol IIIcore alone ($\tau_{on}$ = 15.8 ± 0.9 s, *Figure 4G*). This therefore shows that Pol IIIcore and Pol IV do not form a stable complex on clamp-DNA and that the clamp does not function as a molecular toolbelt, but that the two polymerases compete for the binding of the clamp-DNA in a concentration-dependent manner. This competition is strongly favored towards Pol IIIcore in the presence of the τ clamp loader complex (*Table 1*) as it is directly tethered to Pol IIIcore and ensures that Pol IIIcore is immediately bound upon clamp loading (*Figure 3H*). The presence of the τ clamp loader complex does not alter the frequency of co-localization between Pol IIIcore and Pol IV (*Table 1*). This suggests that during replication, Pol IV may be able to frequently access the clamp-DNA, especially as the estimated cellular concentration of Pol IV is ~10 fold higher than that of Pol IIIcore (*Sutton, 2010*).

Finally, a second *E. coli* translesion polymerase, Pol II (*Paz-Elizur et al., 1996*) also competes with Pol IIIcore for access to the clamp but shows no co-localization with Pol IIIcore (*Figure 4H–I*, *Table 2*), possibly because of its large size (90 kDa *vs.* 40 kDa for Pol IV) that may prevent it from simultaneous binding to the clamp with Pol IIIcore.

## Polymerases compete for binding to the hydrophobic groove of the clamp

To further investigate the competition between the different polymerases and their access to the clamp, we created a series of polymerase mutants. Most clamp interacting proteins, including Pol IIIcore, Pol IV, and Pol II, bind to a hydrophobic groove on the surface of the β-clamp using the canonical sequence Qxx(L/M)xF (*Dalrymple et al., 2001*). The β-clamp is dimeric and thus has two binding grooves. Interestingly, Pol IIIcore contains two β-binding sequences: one in the polymerase subunit α (QADMF, residues 920–924) that is absolutely required for clamp binding (*Dohrmann and McHenry, 2005*), and a second sequence in the exonuclease subunit ε (QTSMAF, residues 182–187) that stabilizes the Pol IIIcore complex and stimulates processive DNA synthesis and exonuclease activity (*Toste Rêgo et al., 2013*; *Jergic et al., 2013*). To demonstrate that the ε β-binding motif also contributes to the lifetime of Pol IIIcore on the clamp, we made two variants of the exonuclease ε, one with a weakened the β-binding motif (QTSMAF to QTSAAA [*Toste Rêgo et al., 2013*]) and one with an enhanced β-binding motif (QTSMAF to QTSLPL [*Fernandez-Leiro et al., 2015*]). Indeed, the lifetime of Pol IIIcore is decreased ~2 fold by the weak β-binding motif, while it is increased ~2

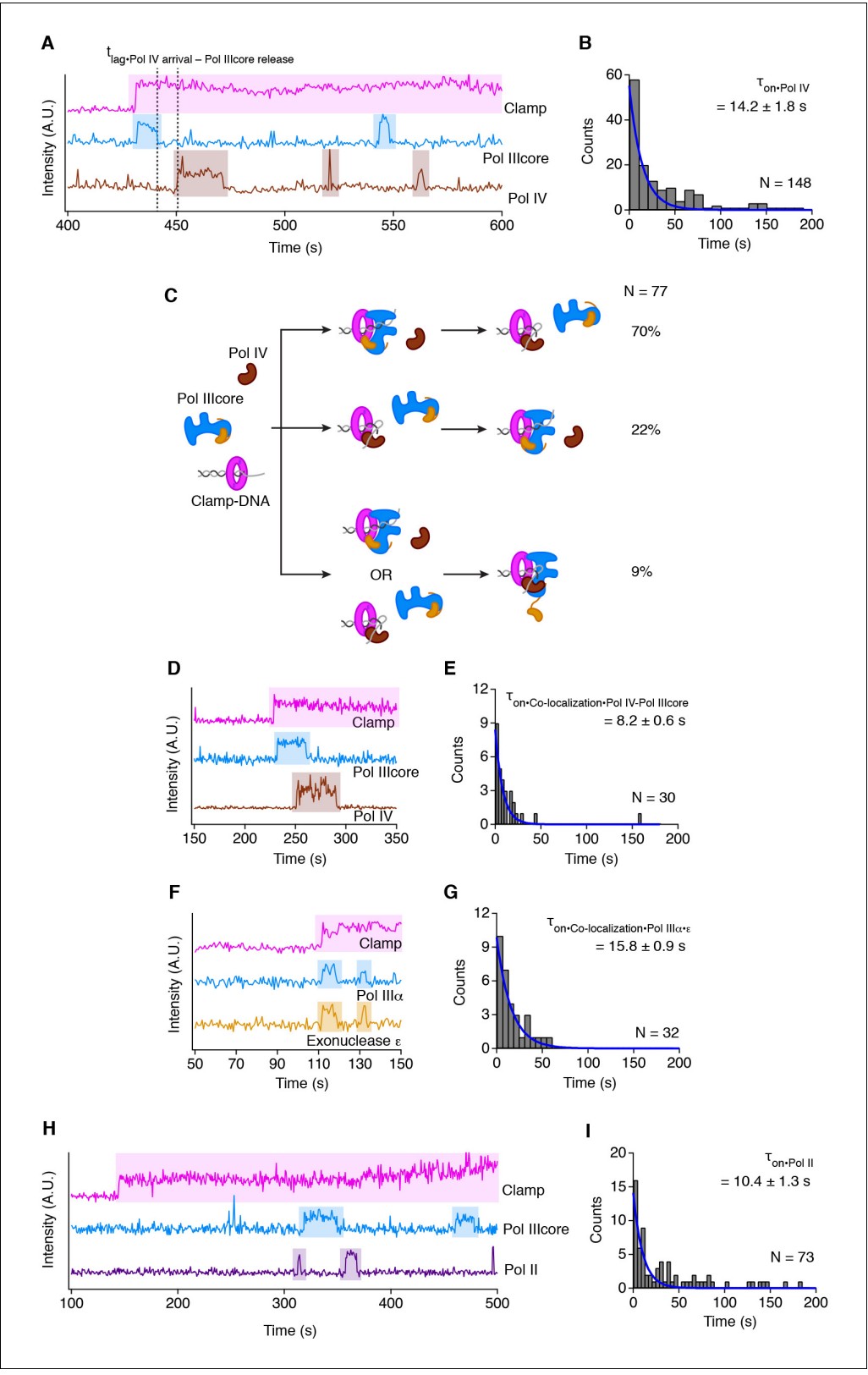

**Figure 4.** The replicative and translesion polymerases compete for binding to clamp-DNA. (**A**) Representative trace showing alternating binding of Pol IIIcore and Pol IV on clamp-DNA. (**B**) Lifetime of Pol IV on clamp-DNA. (**C**) Cartoon showing the frequency of different polymerase switching events. (**D**) Representative trace showing the independent arrival and release of Pol IIIcore and Pol IV on clamp-DNA during co-localization events. (**E**) Lifetime
*Figure 4 continued on next page*

*Figure 4 continued*

of the co-localization of Pol IIIcore on Pol IV on clamp-DNA. (F) Representative trace showing the simultaneous arrival and release of Pol IIIcore α subunit (polymerase) and Pol IIIcore ε subunit (exonuclease) on clamp-DNA. (G) Lifetime of the co-localization of the Pol IIIcore α subunit and Pol IIIcore ε subunit. (H) Representative trace showing alternating binding of Pol IIIcore and Pol II on clamp-DNA. (I) Lifetime of Pol II on clamp-DNA. All values represent mean lifetime ±s.e.m.

DOI: https://doi.org/10.7554/eLife.32177.007

The following figure supplements are available for figure 4:

**Figure supplement 1.** Concentration-dependent competition between Pol IIIcore and Pol IV.
DOI: https://doi.org/10.7554/eLife.32177.008
**Figure supplement 2.** Lifetimes of β-clamp binding mutants of Pol IIIcore, Pol IV and Pol II.
DOI: https://doi.org/10.7554/eLife.32177.009
**Figure supplement 3.** Lesions and mismatches do not affect the lifetime of Pol IIIcore on clamp-DNA.
DOI: https://doi.org/10.7554/eLife.32177.010

fold by the enhanced β-binding sequence (*Table 2*, *Figure 4—figure supplement 2A–B*), showing that Pol III core occupies both binding grooves of the dimeric clamp.

In Pol IV, two β-clamp interacting motifs have been described: a canonical QLVLGL motif (residues 346–351) that binds the hydrophobic groove, and a rim binding sequence (VWP, residues 301–304) that interacts with the side of the clamp (*Becherel et al., 2002*; *Bunting et al., 2003*; *Heltzel et al., 2009*). Mutation of the groove binding motif in Pol IV was reported to inhibit clamp-dependent DNA synthesis, while mutation of the rim contacts resulted in loss of polymerase switching (*Heltzel et al., 2009*). In our experiments, mutation of the groove binding sequence (QLVLGL to QLVAGA, residues 346–351) results in a drastic reduction in the lifetime of binding to the clamp (*Table 2*, *Figure 4—figure supplement 2C*), and consequently no co-localization between Pol IV and Pol IIIcore are observed, even at elevated concentrations of Pol IV (*Table 2*). In contrast mutation of the rim binding motif (VWP to AGA, residues 303–305) shows little effect on the lifetime of Pol IV on the clamp, or on the co-localization frequency with Pol IIIcore (*Table 2*, *Figure 4—figure supplement 2D*). Finally, also mutation of the groove binding motif in Pol II (QLGLF to QLGAA, residues 779–783) leads to a reduced lifetime of binding to the β-clamp (*Table 2*, *Figure 4—figure supplement 2E*).

This therefore shows that all three polymerases compete for the same binding groove on the clamp, and that the isolated polymerases compete with similar lifetimes on the clamp-DNA. This equilibrium is directly influenced by the concentration of the polymerases, or by physically tethering the polymerase to the clamp loader, as is observed for the τ clamp loader complex and Pol IIIcore (*Table 1*, *Figure 3H–J*)

**Table 1.** Competition of Pol IIIcore and Pol IV

| | Concentration (nM) | | Polymerase exchange (%)* | | | Lag time (s)[†] | Lifetime (s)[‡] |
|---|---|---|---|---|---|---|---|
| Competition | Pol IIIcore | Pol IV | III→IV | IV →III | III + IV | III→IV | Pol IIIcore |
| Pol IIIcore - Pol IV | 30 | 6 | 81 | 12 | 7 | 32.5 ± 4.6 | 16.3 ± 1.0 |
| | 30 | 30 | 70 | 22 | 9 | 20.3 ± 3.5 | 15.7 ± 1.1 |
| | 30 | 150 | 63 | 20 | 15 | 5.9 ± 0.5 | 16.6 ± 1.7 |
| | 150 | 150 | 51 | 24 | 26 | 3.5 ± 0.2 | 16.0 ± 0.9 |
| τ-complex[§] - Pol IV | 30[#] | 30 | 95 | 0 | 5 | 11.3 ± 1.3 | 14.8 ± 0.9 |

*Polymerase exchange observed on clamp-DNA showing the exchange from Pol IIIcore to Pol IV (III→IV), Pol IV to Pol IIIcore (IV→III), or co-localization of Pol IIIcore and Pol IV (III + IV).
[†]Time between Pol IIIcore release and Pol IV arrival.
[‡]Lifetime on clamp-DNA.
[§]τ-complex consists of τ clamp loader ($\tau_3\delta_1\delta'_1$) and three Pol IIIcore complexes (α, ε, θ).
[#]Concentration of Pol IIIcore.
DOI: https://doi.org/10.7554/eLife.32177.011

**Table 2.** Lifetime of β-clamp binding mutants of Pol IIIcore, Pol IV and Pol II

| Polymerase | Mutation | Lifetime (s)[†] | Polymerase exchange (%) [*] | | |
|---|---|---|---|---|---|
| | | | III→IV | IV →III | III + IV |
| Pol IIIcore | WT | 15.7 ± 1.1 | 70 | 22 | 9 |
| | ε (β-) | 7.9 ± 1.2 | 70 | 26 | 3 |
| | ε (β+) | 40.2 ± 8.7 | 71 | 24 | 6 |
| Pol IV | WT | 14.2 ± 1.8 | 70 | 22 | 9 |
| | β groove[‡] | 2.7 ± 0.2 | 40 | 60 | 0 |
| | β rim | 14.9 ± 1.7 | 66 | 29 | 5 |
| Polymerase | Mutation | Lifetime (s)[†] | III→II | II →III | III + II |
| Pol II | WT | 10.4 ± 1.3 | 71 | 29 | 0 |
| | β groove | 4.4 ± 0.8 | 63 | 37 | 0 |

[*]Polymerase exchange observed on clamp-DNA showing the exchange from Pol IIIcore to Pol IV or Pol II, Pol IV or Pol II to Pol IIIcore, or co-localization of Pol IIIcore and Pol IV or Pol II.

[†]Lifetime on clamp-DNA.

[‡]The Pol IV β cleft mutant was measured at high concentrations (90nM) in an attempt to catch co-localization events.

DOI: https://doi.org/10.7554/eLife.32177.012

## DNA lesions do not affect the recruitment of translesion polymerases

The apparent lack of organization for the switching of the two polymerases raises the question of how Pol IV is recruited to the site of a lesion. We therefore wondered whether the lifetime of Pol III-core is affected by the nature of the DNA substrate. For this, we compared the lifetimes on three different DNA substrates: a matched, a mismatched, and a substrate containing a $N^2$-furfuryl-dG lesion (*Jarosz et al., 2006*). These three DNA substrates should elicit very different outcomes, i.e. extension on a matched DNA substrate, mismatch removal by the exonuclease ε on a mismatched substrate, or polymerase switching on a DNA lesion. Surprisingly, the lifetime of Pol IIIcore on clamp-DNA is not altered by the presence of either a mismatch or the lesion (*Table 3*, *Figure 4—figure supplement 3A–C*). This indicates that Pol IIIcore does not 'discriminate' between the different DNA substrates. This is also observed in the presence of the two nucleotides dATP and dTTP (0.5 mM each), which give little change in the lifetime of Pol IIIcore on all three DNA substrates (*Table 3*, *Figure 4—figure supplement 3D–F*). Likewise, the exchange of Pol IIIcore to Pol IV is similar on all three substrates, (*Table 3*). Hence, Pol IIIcore dissociation is unaffected by mismatches or DNA lesions and the exchange between Pol IIIcore and Pol IV is not driven by the state of the DNA, but instead is a direct competition between the replicative and translesion polymerases.

**Table 3.** DNA lesion and mismatches do not affect the lifetime of Pol IIIcore on clamp-DNA or its competition with Pol IV

| | Lifetime (s) | | Polymerase exchange[†] (%) | | |
|---|---|---|---|---|---|
| | No dNTP | dATP/dTTP | III → IV | IV → III | III + IV |
| Matched | 15.7 ± 1.1 | 16.1 ± 1.0 | 73 | 14 | 13 |
| Lesion* | 17.6 ± 2.1 | 16.4 ± 1.4 | 58 | 24 | 18 |
| Mismatched* | 19.0 ± 1.4 | 17.5 ± 0.5 | 64 | 23 | 13 |

*Lesion DNA: $N^2$-furfuryl-dG, mismatched DNA: G-T.

[†]Polymerase exchange on observed on clamp-DNA showing the exchange from Pol IIIcore to Pol IV (III→IV), Pol IV to Pol IIIcore (IV→III), or co-localization of Pol IIIcore and Pol IV (III + IV). Exchange rates measured in the absence of nucleotides

DOI: https://doi.org/10.7554/eLife.32177.013

## Discussion

In this work we use co-localization single molecule studies to show that the replication proteins of the clamp loader, clamp, and Pol IIIcore are a highly organized in their sequential actions, ensuring that DNA replication occurs in an efficient manner (*Figure 5*). In contrast, the translesion polymerases appear to gain access to the DNA in a more stochastic, concentration dependent manner. This implies that the cell needs a mechanism in order to tune the outcome of this competition towards

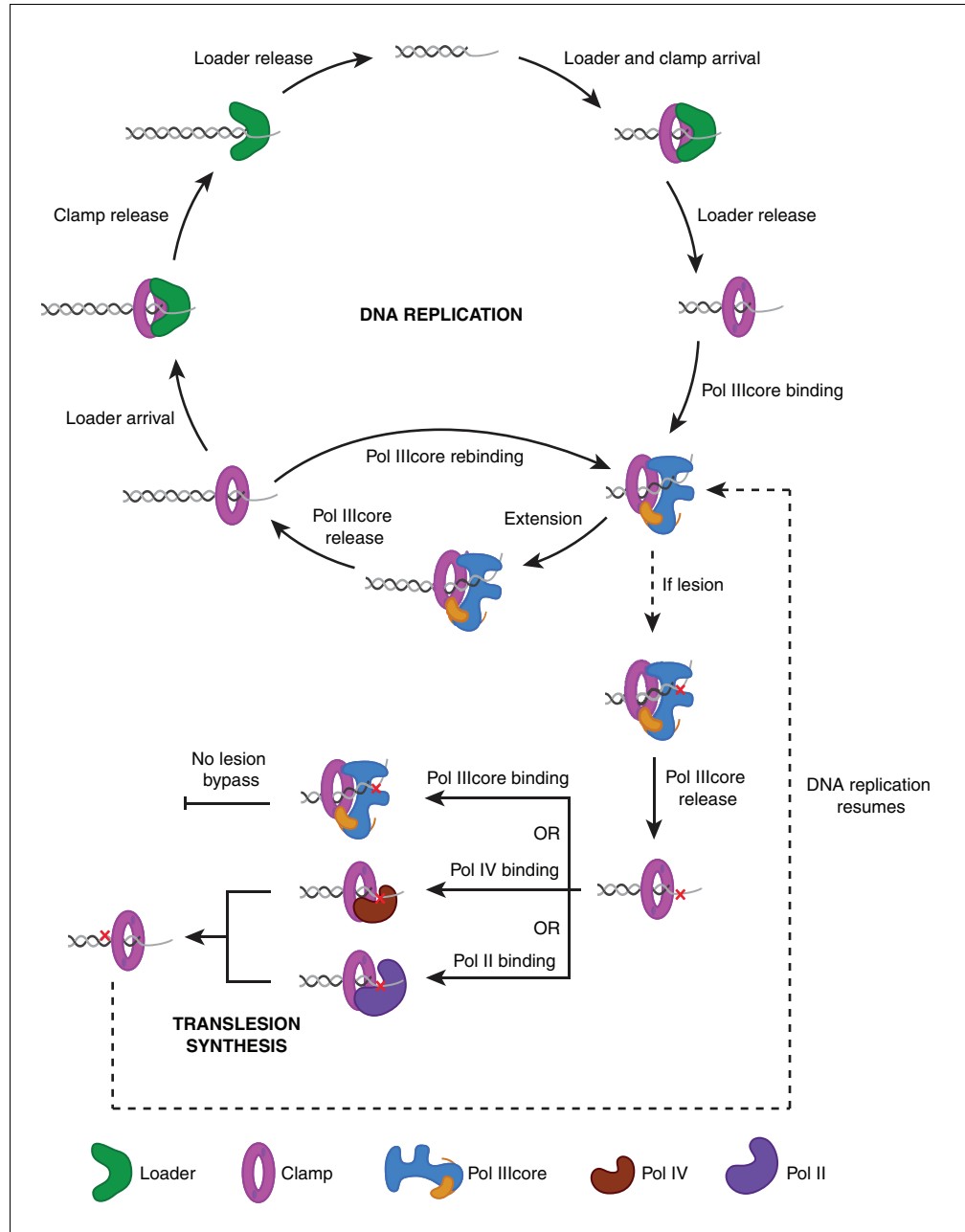

**Figure 5.** A model for DNA replication and translesion synthesis. The DNA replication cycle consists of a sequence of carefully arranged steps of clamp loading, polymerase loading, DNA synthesis, polymerase release, and clamp unloading. In contrast, translesion DNA synthesis over DNA adducts shows no coordinated sequence of events, but is instead a direct competition between the replicative DNA polymerase Pol IIIcore, and the translesion DNA polymerases Pol II and Pol IV. DNA replication will resume once the lesion has been bypassed by one of the translesion DNA polymerases.

DOI: https://doi.org/10.7554/eLife.32177.014

the replicative polymerases during normal DNA synthesis and the translesion polymerases in the presence of DNA damage. Indeed, during DNA synthesis, two or three Pol IIIcores are tethered to the replisome (*McInerney et al., 2007*; *Reyes-Lamothe et al., 2010*), increasing the effective concentration of the replicative polymerase at the replication fork. Importantly, the Pol IIIcores are directly tethered to the clamp loader via the flexible linker of the clamp loader protein τ (*Dallmann et al., 2000*; *Gao and McHenry, 2001*), which ensures that as soon as a clamp is loaded, a Pol IIIcore will associate with the clamp (see *Figure 3H–J*). Secondly, during the SOS-response, a bacterial reaction to DNA damage, the cellular levels of the translesion DNA polymerases Pol II, Pol IV, and Pol V are increased (*Bonner et al., 1988*; *Woodgate and Ennis, 1991*; *Kim et al., 2001*), thus shifting the equilibrium of the polymerase competition in favor the translesion polymerases. Furthermore, it has been proposed that the recombinase RecA may favor the translesion polymerases by having opposing effects on the Pol III and translesion replisomes, albeit by a yet unknown mechanism (*Indiani et al., 2013*).

Such a concentration-dependent exchange between the replicative and translesion polymerases is consistent with the discontinuous DNA synthesis by isolated Pol IIIcore (*Jergic et al., 2013*; *Tanner et al., 2008*), showing that individual Pol IIIcores can exchange. Recent studies (*Lewis et al., 2017*; *Beattie et al., 2017*) have also shown that in the context of the intact replisome, Pol IIIcore exchanges in a concentration-dependent manner during processive DNA replication. Taken together, these studies demonstrate that *E. coli* DNA synthesis is highly dynamic and utilizes a concentration-dependent mechanism to achieve a fine balance between stability and flexibility where the exchange of factors is determined by their availability.

Interestingly, in higher eukaryotes the access of the eukaryotic translesion polymerases to DNA shows a higher degree of coordination through ubiquitination of PCNA and the translesion polymerases (*Chun and Jin, 2010*), as well as formation of polymerase bridges through Rev1 (*Pustovalova et al., 2016*; *Sale, 2013*). However, given that there are up to 15 DNA polymerases to coordinate in eukaryotes (*Plosky and Woodgate, 2004*), the ubiquitination of PCNA and the interactions between the polymerases may not be sufficient to coordinate the specific recruitment of individual translesion polymerases, implying that in eukaryotes too, the translesion process may occur at least in part by a concentration-dependent mechanism, which is supported by studies showing that the intracellular levels of several human translesion polymerases (Pol η, Pol κ, and Pol ι) are increased upon DNA damage (*Zhu et al., 2010*; *Zhu et al., 2012*; *Tomicic et al., 2014*).

# Materials and methods

## Materials
All chemicals were purchased from Sigma-Aldrich (Gillingham, United Kingdom), unless stated otherwise. All chromatography columns were purchased from GE healthcare (Little Chalfont, United Kingdom) .

## Cloning of protein expression vectors
Genes for *E. coli* β (*dnaN*), ε (*dnaQ*), θ (*holE*), γ (*dnaX*) and *Bacillus subtilis* Sfp phosphopantetheinyl transferase were cloned into pET28a vectors, and genes for δ (*holA*) and single cysteine δ' (*holB*) K83C/C217S/C294S(*Goedken et al., 2004*) were cloned into pET3d vectors. The sequence for Pol I (*polA*) Klenow fragment (residues 324–928) was cloned into a pETNKI-His-3C-LIC (*Luna-Vargas et al., 2011*) vector. For labeling purposes the Ybbr sequence DSLEFIASKLA (*Yin et al., 2005*) was added N-terminally to the following proteins during cloning into their respective vectors: Pol IIIα (*dnaE*) was cloned into a pETNKI-His-3C-LIC vector, Pol II (*polB*) and Pol IV (*dinB*) were cloned into pET11 vectors, and the gene for ε (*dnaQ*) was cloned into a pET28a vector. The plasmids for streptavidin 'alive' (biotin-binding) and streptavidin 'dead' (not biotin-binding) (*Howarth et al., 2006*) were generous gifts from M. Howarth (Univ. of Oxford).

Clamp binding mutants of ε, Pol IV, and Pol II were generated through site directed mutagenesis. The following sequences were changed: ε (β-) residues 182–187: QTSMAF to QTSAAA, ε (β+) residues 182–187: QTSMAF to QTSLPL, Pol IV (β groove) residues 346–351: QLVLGL to QLVAGA, Pol IV (β rim) residues 303–305 VWP to AGA, Pol II (β groove), residues 779–783: QLGLF to QLGAA.

## Protein purification

Unless otherwise stated, protein purifications were performed with the following gradients: nickel affinity (25–500 mM Imidazole gradient in the presence of 500 mM NaCl), ion exchange (0–1 M NaCl gradient), hydrophobic interaction chromatography (2–0 M ammonium sulfate gradient) and gel filtration (150 mM NaCl). Pol IIIα-NYbbr was purified in 20 mM Hepes pH 7.5 and 2 mM DTT by nickel affinity, ion exchange and gel filtration. Pol I Klenow fragment was purified in 20 mM Tris pH 8.0 and 2 mM DTT by nickel affinity, anion exchange, hydrophobic interaction chromatography and and gel filtration. Pol IV-NYbbr was purified in 20 mM Hepes pH 7.5 and 2 mM DTT by ion exchange, hydrophobic chromatography and gel filtration. Pol II-NYbbr was purified in 20 mM Tris pH 8.0, 0.5 mM EDTA and 2 mM DTT by two steps of ion exchange separated by a step of hydrophobic chromatography, followed by gel filtration. ε-NYbbr was purified in 25 mM Hepes pH 8.2 and 2 mM DTT by nickel affinity under denaturing conditions (in the presence of 6 M Urea), followed by refolding overnight at 4°C in 25 mM Hepes pH 8.2 and 10 mM DTT and ion exchange. β, ε, θ and Sfp were purified in 20 mM Hepes pH 7.5 and 2 mM DTT by nickel affinity, ion exchange and gel filtration. δ and δ' K83C/C217S/C294S were purified in 50 mM Hepes pH 7.5, 0.1 mM EDTA and 2 mM DTT by hydrophobic chromatography and ion exchange, and γ/τ was purified in the same buffer by nickel affinity chromatography. The $\gamma_3\delta_1\delta'_1$ and $\tau_3\delta_1\delta'_1$ complexes were assembled at a 1:2:1 ratio of γ/τ:δ:δ' and separated by ion exchange. To fully occupy all Pol IIIcore binding site, a 5-fold excess of Pol IIIcore was added to the $\tau_3\delta_1\delta'_1$ complex, and then purified by gel filtration. Streptavidin alive and dead were purified and prepared as described (*Howarth et al., 2006*).

## Protein labeling

Pol IIIα, Pol IV, Pol II and ε exonuclease subunit were enzymatically labeled by Sfp, which conjugates CoA-linked Atto dyes to their N-terminal Ybbr tags (*Yin et al., 2006*). The labeled proteins were Pol IIIα-Atto 488, Pol IV-Atto 565, Pol II-Atto 647N, and ε-Atto 565. The β clamp and Pol I Klenow fragment were labeled on a single cysteine residue with maleimide-Atto647N. The γ-complex (δγ₃δ') was labeled at the δ' subunit with the mutations K83C/C217S/C294S (*Goedken et al., 2004*) with maleimide-Atto 565 before complex assembly. All labeled proteins were purified away from the free dye by gel filtration. The labeling efficiencies of the proteins were determined using the protein and fluorophore absorption ratios, with the free fluorophore absorption at 280 nm subtracted from the protein absorption at 280 nm. The labeling efficiency was 85% for the γ- and τ-complexes, 68% for the β clamp monomer, 100% for Pol I Klenow fragment, 60% for Pol IIIα, 78% for ε, 62% for Pol IV and 66% for Pol II. In addition, the labeling efficiencies for Pol IIIα and ε were verified at the single molecule level by measuring the co-localization frequencies between Pol IIIα and ε on clamp-DNA, where each co-localization event was scored as having labeled Pol IIIα only, labeled ε only, or both labeled proteins. This way, the labeling efficiencies were 67% for Pol IIIα and 71% for ε, which are similar to the efficiencies measured by absorption.

## φX174 primer extension assays

Protein activity was tested using single stranded φX174 phage DNA (New England Biolabs, Hitchin, United Kingdom), primed with a 5' fluorescein labeled primer (sequence: 5' FAM-ACCAACA TAAACATTATTGCCCGGCGTACpG, where lowercase 'p' indicates the non-cleavable phosphothioate bond). Reactions were performed in 20 mM Tris pH 7.5, 2 mM DTT, 50 mM potassium glutamate, 8 mM magnesium acetate, and 0,05 mg/ml BSA. Each reaction contained 5 nM primed φX174 phage DNA, 50 nM β clamp, 10 nM γ clamp loader complex ($\gamma_3\delta_1\delta'_1$), and 30 nM polymerase (Pol IIIcore, Pol II, or Pol IV). Reactions were quenched at 0, 0.5, 1, 2 and 5 min with 75 mM EDTA and 0.6% (W/V) SDS and stored on ice before separated on a alkaline agarose gel (0.8% agarose, 30 mM NaOH, 2 mM EDTA) for 15 hr at 14 V. Gels were scanned at 488 nM using a Amersham Typhoon (GE Healthcare)

## DNA substrates

All DNA oligos were ordered from IDT (Leuven, Belgium), with the exception of the furfuryl-modified oligo that was purchased from Eurogentec (Seraing, Belgium). The following DNA substrates were used. A 33-nt template DNA: 5' Bio-CATAATATCCATGCTTCACC[amino-dT]TCATCCAAATCC for the matched and mismatched substrates or 5' Bio-CATAA[N2-furfuryl-dG]ATCCATGCTTCACC

[amino-dT]TCATCCAAATCC for the lesion substrate. A 27-nt primer DNA: 5' Bio-GGATTTGGA TGAAGGTGAAGCATGGApT for the matched and lesion substrates (where lowercase 'p' indicates the non-cleavable phosphothioate bond) or a 25-nt 5' Bio-GGATTTGGATGAAGGTGAAGCATGpT for the mismatched substrate. The template DNA was labeled on the internal amino-modified thymine with NHS-Atto 488 and purified away from the free dye by gel filtration. The labeled template DNA was subsequently bound to monovalent streptavidin and purified by gel filtration, before annealing to the primer DNA and binding to the glass cover slip for imaging. See *Figure 1A* for cartoon representation of final DNA substrate.

### Preparation of slides

Glass slides and cover slips were washed in 3 M NaOH and Piranha solution (3:2 concentrated sulfuric acid: 30% hydrogen peroxide) and then silanized and pegylated essentially as described in (*Ha et al., 2002*). The imaging chamber (15 μL) was assembled by creating a sandwich between the cover slip and glass slide using double adhesive tape, and it was further passivated using 4 mg/ml PLL-PEG (SuSoS, Dübendorf, Switzerland), 1% (W/V) pluronic F127 (Sigma) and 10 mg/ml BSA (New England Biolabs). Streptavidin (1 mg/ml) (New England Biolabs) was added last to bind the biotin-DNA.

### CoSMoS microscopy

All single-molecule measurements were performed at 23 (±1) °C in 20 mM Tris-HCl pH 7.5, 50 mM potassium glutamate, 8 mM MgCl$_2$, 4% glycerol, 2 mM DTT, 0.1% Tween20 and 1 mM Trolox. The protein concentrations used were: 15 nM β$_2$ clamp; 15 nM γ$_3$δ$_1$δ'$_1$ complex; 30 nM or 150 nM Pol III-core; 6 nM, 30 nM, or 150 nM Pol IV; and 30 nM Pol II. Movies were acquired using Micromanager software on a Nikon (Kingston Upon Thames, United Kingdom) Eclipse Ti-E microscope with ApoTirf 100X/1.49 Oil, 0.13–0.20 WD 0.12 objective. The lasers used were 150 mW 488 nm, 150 mW 561 nm (both Coherent Sapphire Ely, United Kingdom) and 100 mW 638 nm (Coherent Cube) controlled by an acousto-optic tunable filter (Gooch and Housego, Ilminster, United Kingdom). Movies were acquired on an Andor (Belfast, United Kingdom) iXon (EM) + CCD camera at 20–40 mW laser power and an exposure of 50 millisecond (ms) per frame for a thousand frames with rapid alternation between the three laser channels (180 ms/change). Thus the frame rate is 660 ms for a 3-color experiment, 440 ms for a 2-color experiment, and 86 ms for a single color experiment (as there is no need to change the laser channels). The total duration of the movies for a 3-color experiment was 650–670 s, which gives an acquisition rate of 220 milliseconds per frame when the lag in the filter wheel was taken into consideration. Each field of view (54 μm × 54 μm, pixel size 105 × 105 nm) had an average density of 5900 DNA molecules of which 700–900 molecules were well separated and picked for analysis. The DNA-Atto 488 signal was bleached in order for this channel to be re-used for Pol IIIα-Atto 488.

### Data analysis

The acquired movies were fed into the Imscroll analysis GUI developed by Jeff Gelles and Larry Friedman (*Friedman and Gelles, 2015*) to find the individual landing events and their dwell times. Histograms were plotted in Igor Pro and the data was fitted with an one-parameter exponential using a bin size ~1/2 the lifetime of the molecule. The decay constant τ represents the mean dwell time and the error in τ represents the error in the mean dwell time.

## Acknowledgements

We would like to thank Andrew Carter and Nick Barry for help with TIRF microscopy, Jeff Gelles and Larry Friedman for help with CoSMoS data analysis, and David Rueda for helpful suggestions. This work was supported by the UK Medical Research Council through grant U105197143 to MHL.

## Additional information

### Funding

| Funder | Grant reference number | Author |
|---|---|---|
| Medical Research Council | U105197143 | Gengjing Zhao<br>Emma S Gleave<br>Meindert Hugo Lamers |

The funders had no role in study design, data collection and interpretation, or the decision to submit the work for publication.

### Author contributions

Gengjing Zhao, Data curation, Formal analysis, Investigation, Writing—original draft, Writing—review and editing; Emma S Gleave, Data curation, Writing—review and editing; Meindert Hugo Lamers, Formal analysis, Supervision, Validation, Writing—original draft, Writing—review and editing

### Author ORCIDs

Meindert Hugo Lamers (iD) http://orcid.org/0000-0002-4205-1338

### Decision letter and Author response

Decision letter https://doi.org/10.7554/eLife.32177.019
Author response https://doi.org/10.7554/eLife.32177.020

## Additional files

### Supplementary files

• Source data 1. Source data file with raw data for *Figures 2–4* and figure supplements.
DOI: https://doi.org/10.7554/eLife.32177.015

• Transparent reporting form
DOI: https://doi.org/10.7554/eLife.32177.016

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
