## [Decision Letter]

[Editors’ note: a previous version of this study was rejected after peer review, but the authors submitted for reconsideration. The first decision letter after peer review is shown below.]

Thank you for submitting your work entitled "Single-molecule studies contrast ordered DNA replication with chaotic translesion synthesis" for consideration by *eLife*. Your article has been reviewed by two peer reviewers, and the evaluation has been overseen by a Reviewing Editor and a Senior Editor. The reviewers have opted to remain anonymous.

Our decision has been reached after consultation between the reviewers. Based on these discussions and the individual reviews below, we regret to inform you that your work will not be considered further for publication in *eLife*.

The reviewers agreed that your approach is novel and the messages touch about timely topics of broad interest. However, three major issues were raised. (1) Controls showing that the labeled components are functional in DNA synthesis are missing. (2) Bolstering their claims by adding much more data and analysis, instead of just showing one or few example time traces, is also a must. (3) Many more additional experiments are needed, for example including clamp loader that bridges pol III and clamp, and measurements that demonstrate the necessity of clamp interaction.

Your paper is being rejected because we do not feel that you would be able to perform all requested additional experiments (most critically point 3 above) that the reviewers found essential in two months, which is the time scale of additional work that *eLife* considers reasonable. However, if you are able to perform key experiments requested, I will be happy to consult the same reviewers with the new submission.

*Reviewer #1:*

Zhao et al. investigate the binding of *E. coli* DNA polymerases to the sliding clamp using single-molecule co-localization microscopy. The authors argue that replicative and translesion polymerase do not simultaneously interact with the clamp (toolbelt model), but exchange on the clamp in a concentration-dependent manner. This manuscript addresses a critical question in the field, albeit with a rather limited scope. Most importantly, clamp loader (with and without τ) is omitted from the polymerase co-localization experiments. Τ bridges clamp and polIII, and may further stabilize the "dual-polymerase" clamp complex in vivo. This will be a critical addition to the manuscript. Second, there are no biochemical controls indicating activity of fluorescently tagged enzymes. Statistical analysis is also lacking and in a few cases the key results are argued based on a single CosMoS trace. Overall, the manuscript feels more like the start of an interesting paper than a significant contribution to the field. I recommend that the authors make the following major edits to be considered for publication in *eLife*.

1) Include biochemical controls to confirm that clamp loader and polymerases are not perturbed by the fluorescent tag.

2) Include experiments with clamp loader, pol III and pol IV. Compare clamp loader with and without τ. Are more "dual polymerase" species seen with τ? There is growing evidence for multi-τ clamp loaders in vivo (from Sherratt lab and others).

3) Figure 1: Clamp on-rates (~0.6 s) are limited by the experimental resolution (0.6 s / frame). The underlying rates may be significantly faster; the authors may be measuring slow events. Time resolution needs to be increased to make any claims here.

4) Once a clamp is loaded, clamp loader can catalyze unloading (e.g., Figure 1). Does every clamp loader binding event result in clamp unloading (as suggested in Figure 1)? Clamp unloading probabilities should be characterized and reported. If unloading is rare, more representative traces shown in Figure 1.

5) In the experiments detailing matched, mismatched, and lesion template, the authors investigate the lifetime without dNTPs and with dCTP/dGTP. If I understood the oligo table correctly, the next nucleotide to be incorporated is dATP or dTTP. Including these nucleotides may show changes in how the polymerase interacts with the templates.

6) Do changes in competing polymerase (Pol II or Pol IV) concentration affect the lifetime of Pol III core on the clamp? This has been suggested previously (PMCID: PMC4040570).

7) Does an increase in polymerase concentration increase the co-association of Pol III core and Pol IV?

8) What are the relevant concentrations of these polymerases in cells before and after SOS induction? These concentration ranges (or at least the relative polymerase ratios) should be investigated here.

9) A separation-of-function Pol III mutant that lacks interaction with the clamp will be useful to confirm that specific interactions are responsible for the increase in lifetime.

10) Controls with weakened clamp-pol IV interactions will also confirm the observed lifetimes and polymerase exchange dynamics (e.g., PMCID: PMC4040570).

11) Authors make a sweeping claim that all three polymerases "compete equally" for the clamp based on a single trace (Figure 3). This is a shaky and unclear claim. Elaborate with more statistical analysis, and with wt and mutant proteins that weaken interactions.

*Reviewer #2:*

This paper uses co-localization microscopy with labeled DNA polymerases to determine the sequence of events of the binding of components of the replisome and different DNA polymerases to DNA. Clamp loading and unloading was observed, as well as association times of pol IIIcore and pol IV or pol II, which supports a model of stochastic binding of pol IV or pol II compared to a more ordered assembly of pol III components.

A major question is: are the tagged DNA polymerases active? The clamp loader and clamp clearly are competent for loading, but polymerization activity (and proofreading) by the tagged polymerases should be demonstrated.

"The shorter lifetimes recorded for the clamp are not caused by early release […] are instead due to the late loading of the clamps during the course of the experiment." Is this because this is observed near the end of the acquisition time? This should be explained.

Figure 2, were these experiments carried out with labeled α or labeled ε subunit? This question holds in every instance that specifies only "pol III core".

Subsection “Pol IIIcore has an intrinsic lifetime on DNA that is independent of its activity”, last paragraph and Table 1, what is the concentration of dGTP and dCTP used in these experiments?

"Pol IIIcore and Pol IV do not form a stable complex on clamp-DNA […] and there is a significant time lag between the release of Pol IIIcore and Pol IV arrival" However, it appears that sometimes Pol IIIcore and Pol IV do stably associate on the clamp (Figure 3 shows a significant amount of time in which the two pols are co-localized). And, top "rare co-localization events." How rare are these? Table 1 shows this to be 13-18% of total events, but what about the time of co-localization? It would seem prudent to include the co-localized times in the analysis as well and present those results here. In the representative trace shown, the co-localized time appears to exceed most of the association lifetimes of either polymerase alone on DNA. It would be of interest to conduct this experiment without β clamp. Also, does the percentage of co-localization change with nucleotides?

Subsection “Protein purification”, more information and/or references should be given describing the protein purification, including buffer compositions.

Subsection “CoSMoS microscopy”, the 488 label on DNA was photobleached before imaging pol IIIcore-α-488. What controls were done to ensure the photobleaching doesn't damage the DNA?

[Editors’ note: what now follows is the decision letter after the authors submitted for further consideration.]

Thank you for submitting your article "Single-molecule studies contrast ordered DNA replication with chaotic translesion synthesis" for consideration by *eLife*. Your article has been reviewed by two peer reviewers, and the evaluation has been overseen by a Reviewing Editor and Michael Marletta as the Senior Editor. The reviewers have opted to remain anonymous.

Your revised manuscript has been seen by the original reviewers. They generally agreed that the manuscript has improved significantly but made several important suggestions that should be helpful in improving rigor and clarity. Please revise your manuscript accordingly. No new experiments are required.

*Reviewer #2:*

This version of the manuscript is improved, but there are still some areas to address.

In the response, the authors state that the only report of polymerase concentrations in *E. coli* is Sutton 2009 BBA, which is a review article. This paper cites four other primary research articles, where the levels of polymerases are actually measured. I understand the desire to keep papers a reasonable length, and so cite a review when appropriate, but it is just not true that Sutton 2009 is the only report of polymerase levels. In addition, if one reads this paper and the cited references, it becomes clear that the levels of pol IV are well above those of pol III even without SOS induction, and therefore statements about "low intracellular levels" of pol IV under "normal" (non-SOS induced?) conditions (Discussion, first paragraph) need to be revisited.

In the first paragraph of the Introduction: "When the errors occur on the template DNA strand, in the form of DNA lesions […]" is misleading. DNA lesions that arise from chemicals, etc., are not replication errors. The wording should be revised.

In the Introduction, studies supporting the toolbelt model "all […] use bulk studies that due to the asynchronous nature cannot separate out the sequential steps." While this statement is true, it seems to ignore the fact that steady-state FRET experiments support the idea that two pols can bind β simultaneously (Indiani 2005 Mol Cell).

In the last paragraph of the Discussion, ordered vs. stochastic, the manuscript doesn't seem to be entirely convincing on this point. The exchange between two different pol III cores (two different labels) should be examined as a direct comparison to the other two-polymerase experiments. Either the experiments can be conducted to or it would be sufficient to revise the discussion, the general interpretation of experiments, and to change the title of manuscript.

Table 1 and Table 2 are confusing. Why is pol II in Table 1? In Table 2, it is not clear what is being compared in each case. The last column gives co-localization, and the heading is "III+IV/II"; which is reported in the values for pol III (first 4 entries), pol II or pol IV?

---

## [Author Response]

[Editors’ note: the author responses to the first round of peer review follow.]

Reviewer #1:Summary:Zhao et al. investigate the binding of E. coli DNA polymerases to the sliding clamp using single-molecule co-localization microscopy. The authors argue that replicative and translesion polymerase do not simultaneously interact with the clamp (toolbelt model), but exchange on the clamp in a concentration-dependent manner. This manuscript addresses a critical question in the field, albeit with a rather limited scope. Most importantly, clamp loader (with and without τ) is omitted from the polymerase co-localization experiments. Τ bridges clamp and polIII, and may further stabilize the "dual-polymerase" clamp complex in vivo. This will be a critical addition to the manuscript.

In the previous version of the manuscript, all experiments were performed in the presence of the γ clamp loader complex (i.e. γ_3_δδ', without τ), with the exception of the clamp unloading and the three-polymerase experiment, which were performed after washing away the clamp loader.

In the current version we have now also included the τ clamp loader complex (τ_3_δδ'), as shown in Figure 3. In summary, when using the τ clamp loader complex that is pre-loaded with Pol IIIcore, we find that the clamp loader, Pol IIIcore, and clamp arrive at the DNA together. Unlike the γ clamp loader that leaves immediately after clamp loading, the τ complex remains present as it is attached to Pol IIIcore that takes its place at the clamp.

The lifetime of Pol IIIcore is not altered in the presence of τ. As the τ clamp loader and Pol IIIcore arrive together, there is a dramatic shift in the competition between Pol IIIcore and Pol IV in favour of Pol III, as shown in the table below (Author response Table 1). The percentage of co-localization events between Pol IIIcore and Pol IV on the other hand is not much affected.

Concentration (nM) Polymerase exchange (%) Lifetime

**Author response table 1. resptable1:** Competition between polymerases clamp-DNA.

DNA	Concentration (nM)	Polymerase exchange (%)	Lifetime
Competition	Pol IIIcore	Pol IV/II	III→IV	IV→III	III+IV	Pol IIIcore
γ complex Pol IIIcore – Pol IV	30	30	70	22	9	15.7 ± 1.1
τ complex Pol IIIcore – Pol IV	30	30	95	0	5	13.5 ± 1.7

Second, there are no biochemical controls indicating activity of fluorescently tagged enzymes.

We have now included bulk biochemical experiments, a 5.4 kb ϕX174 primer extension assay, and show that the labelled proteins (i.e. clamp, clamp loader, pol IIIcore, Pol IV, and Pol II) are unaltered in DNA synthesis when compared to the unlabeled proteins (see Figure 1—figure supplement 1).

Statistical analysis is also lacking and in a few cases the key results are argued based on a single CosMoS trace.

We now have included the statistical analysis for all experiments throughout the figures and figure supplements.

Overall, the manuscript feels more like the start of an interesting paper than a significant contribution to the field. I recommend that the authors make the following major edits to be considered for publication in eLife.1) Include biochemical controls to confirm that clamp loader and polymerases are not perturbed by the fluorescent tag.

As explained above, we have now included primer extension assays using a primed, 5.4 kb single stranded ϕX174 phage DNA that requires clamp loading for processive DNA synthesis. The results show that the labeled proteins retain wild-type activities (see Figure 1—figure supplement 1).

2) Include experiments with clamp loader, pol III and pol IV. Compare clamp loader with and without τ. Are more "dual polymerase" species seen with τ? There is growing evidence for multi-τ clamp loaders in vivo (from Sherratt lab and others).

As discussed above (see "Summary: Most importantly"), these experiments have now been included in the revised manuscript. We have compared the clamp loader with and without τ. We are not sure what the reviewer means with "dual polymerase species". If it is referring to co-localization between Pol IIIcore and Pol IV, then no, we do not observe increased colocalization in the presence of τ Table 1 above). If the reviewers refer to multiple Pol IIIcores bound to one τ clamp loader complex, then yes, we make the τ-complex (τ_3_δ_1_δ'_1_) fully loaded with three Pol IIIcores.

3) Figure 1: Clamp on-rates (~0.6 s) are limited by the experimental resolution (0.6 s / frame). The underlying rates may be significantly faster; the authors may be measuring slow events. Time resolution needs to be increased to make any claims here.

We have repeated the clamp loader experiments using a single fluorescent channel only, thus reducing the frame rate to 0.086 s/frame. At this frame rate, the lifetime of the clamp loader on DNA in the presence or absence of the clamp is 0.41 ± 0.01 s and 1.20 ± 0.05 s, respectively. This is now shown in Figure 2 and discussed in the first paragraph of the subsection “Clamp loading and unloading are distinct processes”.

4) Once a clamp is loaded, clamp loader can catalyze unloading (e.g., Figure 1). Does every clamp loader binding event result in clamp unloading (as suggested in Figure 1)? Clamp unloading probabilities should be characterized and reported. If unloading is rare, more representative traces shown in Figure 1.

We now measured the success rate of unloading, which is ~50%. This is now shown in Figure 2—figure supplement 1 discussed in the last paragraph of the subsection “Clamp loading and unloading are distinct processes”.

5) In the experiments detailing matched, mismatched, and lesion template, the authors investigate the lifetime without dNTPs and with dCTP/dGTP. If I understood the oligo table correctly, the next nucleotide to be incorporated is dATP or dTTP. Including these nucleotides may show changes in how the polymerase interacts with the templates.

In the initial manuscript, we chose not to use as dATP and dTTP as their incorporation would shorten the ssDNA overhang by 5 nucleotides and potentially change the interaction with the polymerase (in many crystal structures of polymerase-DNA complexes, the polymerase binds to 3-4 nucleotides of the single stranded overhang of the template strand. Our DNA substrate has a 6 nt overhang). Moreover, Pol IIIcore synthesizes DNA at speeds of 1000 nt/s, so the addition of 5 nucleotides could only add 5 milliseconds to the ~16 second lifetime of Pol IIIcore on clamp-DNA. Finally, the end product of incorporation of the five nucleotides would be the same as the starting DNA substrate with dCTP and dGTP, i.e. a primer that cannot be extended. Nonetheless, we have performed the idling experiment with dATP and dTTP and found that as for dCTP and dGTP, the lifetime of Pol IIIcore is comparable on all three DNA substrates (matched, mismatched, and lesion). This is now shown in Table 3 and discussed in the subsection “DNA lesions do not affect the recruitment of translesion polymerases”.

6) Do changes in competing polymerase (Pol II or Pol IV) concentration affect the lifetime of Pol III core on the clamp? This has been suggested previously (PMCID: PMC4040570).

We have now measured different concentrations of Pol IV (6-150 nM) and do not see an effect on the lifetime of Pol IIIcore. We do see a clear effect on the polymerase switching (III>IV vs IV->III), a decrease in the lag time between binding events, and an increase of colocalizations of Pol IIIcore and Pol IV with increased protein concentrations. This is now shown in Table 1 and discussed in the first paragraph of the subsection “Pol IIIcore and Pol IV alternate binding to clamp-DNA”.

7) Does an increase in polymerase concentration increase the co-association of Pol III core and Pol IV?

Yes it does. This is now shown in Table 1 and discussed in the second paragraph of the subsection “Pol IIIcore and Pol IV alternate binding to clamp-DNA”.

8) What are the relevant concentrations of these polymerases in cells before and after SOS induction? These concentration ranges (or at least the relative polymerase ratios) should be investigated here.

To our knowledge, there is currently only one report of the cellular concentration of the different polymerases: Sutton et al. 2009 (PMID 19540941). Here the cellular concentrations of *E. coli* Pol IIIcore and Pol IV are 20 and 330 nM respectively, and 20 and 3300 nM after SOS induction. In our experiments, we have measured Pol IIIcore and Pol IV at 6, 30, and 150 nM (Table 1). (Measuring higher protein concentrations results in increased background and non-specific binding to the glass slide.) At the measured concentrations it is clear that Pol IV can frequently switch places and/or co-localize with Pol IIIcore. Addition of τ favors the binding of Pol IIIcore to the DNA, although the frequency of co-localization with Pol IV is comparable to that of the isolated Pol IIIcore. In the cell, Pol IIIcore is connected to the rest of the replisome, implying that the local effective concentration will favor Pol IIIcore binding even more. This is now shown in Table 1 and discussed in the second paragraph of the subsection “Pol IIIcore and Pol IV alternate binding to clamp-DNA”.

9) A separation-of-function Pol III mutant that lacks interaction with the clamp will be useful to confirm that specific interactions are responsible for the increase in lifetime.

The clamp interaction motif of the polymerase subunit α of Pol IIIcore is essential for clamp binding and DNA synthesis (Dohrman 2005, PMID 15923012). Instead we have made two variants of the clamp binding motif in the exonuclease subunit ε of Pol IIIcore; one to decrease clamp binding and one to increase clamp binding. As expected, the reduced clamp binder of ε results in shorter lifetimes of Pol IIIcore on clamp-DNA, and the improved clamp binder of ε gives longer lifetimes of Pol IIIcore on clamp-DNA. This is now shown in Table 2 and discussed in the first paragraph of the subsection “Polymerases compete for binding to the hydrophobic groove of the clamp”.

10) Controls with weakened clamp-pol IV interactions will also confirm the observed lifetimes and polymerase exchange dynamics (e.g., PMCID: PMC4040570).

We have made two mutant version of Pol IV: one in the canonical groove binding sequence, and one in the rim binding sequence (see Heltzel 2009, PMID 19617571). We find that mutation of the groove binding sequence strongly reduces Pol IV binding to the clamp and prevents co-localization. Mutation of the rim binding sequence of Pol IV had no observable effect. This is shown in Table 2 and discussed in the second paragraph of the subsection “Polymerases compete for binding to the hydrophobic groove of the clamp”.

11) Authors make a sweeping claim that all three polymerases "compete equally" for the clamp based on a single trace (Figure 3). This is a shaky and unclear claim. Elaborate with more statistical analysis, and with wt and mutant proteins that weaken interactions.

We have replaced this statement with: "This therefore shows that all three polymerases compete for the same binding groove on the clamp, and that the isolated polymerases compete with similar lifetimes on the clamp-DNA."

Reviewer #2:[…] A major question is: are the tagged DNA polymerases active? The clamp loader and clamp clearly are competent for loading, but polymerization activity (and proofreading) by the tagged polymerases should be demonstrated.

We have now included primer extension assays using a primed, 5.4 kb single stranded ϕX174 phage DNA that requires clamp loading for processive DNA synthesis. The results show that the labeled proteins retain wild-type activities. This is shown Figure 1—figure supplement 1, and discussed in the subsection “Preparation of DNA substrates and fluorescently labeled proteins”.

"The shorter lifetimes recorded for the clamp are not caused by early release […] are instead due to the late loading of the clamps during the course of the experiment." Is this because this is observed near the end of the acquisition time? This should be explained.

We have now measured the clamp lifetime in a different manner. We first loaded the clamp using the clamp loader, then washed away the loader and unbound clamps and then started data collection using 10 seconds intervals between measurements to avoid bleaching of the fluorophore. This resulted in a measured lifetime of ~23 minutes, not including the loading and washing time. This is now shown in Figure 2 and discussed in the first paragraph of the subsection “Clamp loading and unloading are distinct processes”.

Figure 2, were these experiments carried out with labeled α or labeled ε subunit? This question holds in every instance that specifies only "pol III core".

All experiments with Pol IIIcore were performed with a labeled α subunit. This is now stated in the subsection "Preparation of DNA substrates and fluorescently labeled proteins". Only for the co-localization of α and ε we also used a labeled ε subunit (Figure 4).

Subsection “Pol IIIcore has an intrinsic lifetime on DNA that is independent of its activity”, last paragraph and Table 1, what is the concentration of dGTP and dCTP used in these experiments?

0.5 mM for each nucleotide. This is now stated in the text (subsection “Pol IIIcore has an intrinsic lifetime on DNA that is independent of its activity”, second paragraph; subsection “Pol IIIcore and Pol IV alternate binding to clamp-DNA”, first paragraph; subsection “DNA lesions do not affect the recruitment of translesion polymerases”)

"Pol IIIcore and Pol IV do not form a stable complex on clamp-DNA […] and there is a significant time lag between the release of Pol IIIcore and Pol IV arrival" However, it appears that sometimes Pol IIIcore and Pol IV do stably associate on the clamp (Figure 3 shows a significant amount of time in which the two pols are co-localized). And, top "rare co-localization events." How rare are these? Table 1 shows this to be 13-18% of total events, but what about the time of co-localization? It would seem prudent to include the co-localized times in the analysis as well and present those results here.

We have extended our analysis of the co-localization events by testing different concentrations of Pol IIIcore and Pol IV (see Table 1). In the table we report the percentage of co-localization events, that increases from 9% at low protein concentrations to 26% at high protein concentrations. The measured lifetime is ~8 seconds. This is now discussed shown in Table 1 and discussed in the second paragraph of the subsection 2Pol IIIcore and Pol IV alternate binding to clamp-DNA”.

In the representative trace shown, the co-localized time appears to exceed most of the association lifetimes of either polymerase alone on DNA. It would be of interest to conduct this experiment without β clamp.

In the absence of the clamp, Pol IIIcore binds very transiently to the DNA (τ_on_=<0.086 s). This therefore makes it highly unlikely for any co-localization between Pol IIIcore and Pol IV to occur. For example, a β-clamp mutant of pol IV that has a reduced lifetime of 2.7 s (compared to 15.4 s for wild type Pol IV) shows no co-localization with Pol IIIcore on clampDNA even at elevated concentrations (see Table 2).

Also, does the percentage of co-localization change with nucleotides?

No it does not. This is now mentioned in thein the second paragraph of the subsection 2Pol IIIcore and Pol IV alternate binding to clamp-DNA”.

Subsection “Protein purification”, more information and/or references should be given describing the protein purification, including buffer compositions.

We have now expanded the information regarding the protein purification (see Materials and methods, subsection “Protein purification”).

Subsection “CoSMoS microscopy”, the 488 label on DNA was photobleached before imaging pol IIIcore-α-488. What controls were done to ensure the photobleaching doesn't damage the DNA?

We have measured the lifetime of Pol IIIcore on clamp-DNA before and after photobleaching of DNA and find there is no difference. This is now shown in Figure 1—figure supplement 1 and mentioned in the subsection “Preparation of DNA substrates and fluorescently labeled proteins”.

Furthermore, we bleach the Atto 488 dye at its absorption peak at 488 nm. The absorption of DNA drops sharply to zero a >300 nm, so the amount of energy absorbed by the DNA at 488 nm is extremely low. Consequently, UV induced damage is negligible at > 300 nm (see Besaratinia et al., 2011, PMID 21613571).

[Editors' note: the author responses to the re-review follow.]

Reviewer #2:[…] In the response, the authors state that the only report of polymerase concentrations in E. coli is Sutton 2009 BBA, which is a review article. This paper cites four other primary research articles, where the levels of polymerases are actually measured. I understand the desire to keep papers a reasonable length, and so cite a review when appropriate, but it is just not true that Sutton 2009 is the only report of polymerase levels. In addition, if one reads this paper and the cited references, it becomes clear that the levels of pol IV are well above those of pol III even without SOS induction, and therefore statements about "low intracellular levels" of pol IV under "normal" (non-SOS induced?) conditions (Discussion, first paragraph) need to be revisited.

We now cite the original references (Bonner et al., 1988, Woodgate and Ennis, 1991, Kim et al., 2001) that report the increased expression of Pol II, Pol IV, and Pol V during the SOS response (see: Discussion, first paragraph). Three other references listed in the Sutton 2009 paper (#36 Qiu '97, #37 Bonner '90, #146 McHenry '03) actually do not report protein levels and are therefore not included in our revised manuscript. In addition, we adjusted our wording regarding 'normal' and 'low' levels of pol IV and changed to "Secondly, during the SOS-response, a bacterial reaction to DNA damage, the cellular levels of the translesion DNA polymerases Pol II, Pol IV, and Pol V are increased [Bonner et al., 1988; Woodgate and Ennis, 1991; Kim et al., 2001], thus shifting the equilibrium of the polymerase competition in favor the translesion polymerases."

In the first paragraph of the Introduction: "When the errors occur on the template DNA strand, in the form of DNA lesions […]" is misleading. DNA lesions that arise from chemicals, etc., are not replication errors. The wording should be revised.

We have modified the text to be more clear: "In addition, rare nucleotide mis-incorporations into the primer strand prevent further DNA synthesis and a 3'-5' exonuclease is required to remove the misincorporated nucleotides [Kunkel, 2004]. In contrast, when the polymerase encounters a lesion on the template strand in the form of modified base caused by diverse chemicals, reactive oxygen species, or UV light [Liu et al., 2016; Lindahl, 1996], the high-fidelity replicative DNA polymerases are stalled."

In the Introduction, studies supporting the toolbelt model "all […] use bulk studies that due to the asynchronous nature cannot separate out the sequential steps." While this statement is true, it seems to ignore the fact that steady-state FRET experiments support the idea that two pols can bind β simultaneously (Indiani 2005 Mol Cell).

We have now included a statement about the FRET experiments of the Indiani 2005 paper: "The toolbelt model, which was originally based on steady-state Förster Energy Resonance Transfer (FRET) experiments that showed the simultaneous binding of the replicative and translesion polymerases to the β-clamp, has found support.…"

In the last paragraph of the Discussion, ordered vs. stochastic, the manuscript doesn't seem to be entirely convincing on this point. The exchange between two different pol III cores (two different labels) should be examined as a direct comparison to the other two-polymerase experiments. Either the experiments can be conducted to or it would be sufficient to revise the discussion, the general interpretation of experiments, and to change the title of manuscript.

As we show in Figure 2 and Figure 3, the loading of the clamp and pol IIIcore occur in a well-defined and ordered sequence. In contrast, as shown in Figure 4, the exchange between Pol IIIcore and the translesion polymerase Pol II and Pol IV does not follow any particular sequence, and any possible exchange between the polymerase are observed. Hence, the polymerase exchange follows a random probability, which is the definition of stochastic. We therefore stand by our distinction between ordered replication and stochastic translesion synthesis. However, we have replaced "chaotic" with "stochastic" in the title, and have toned down the Discussion.

Table 1 and Table 2 are confusing. Why is pol II in Table 1? In Table 2, it is not clear what is being compared in each case. The last column gives co-localization, and the heading is "III+IV/II"; which is reported in the values for pol III (first 4 entries), pol II or pol IV?

We have adjusted Table 1 and Table 2 to remove the confusion. Pol II has been removed from the column heading in Table 1, and we have added a footnote to explain the polymerase exchange ("Polymerase exchange observed on clamp-DNA showing the exchange from Pol IIIcore to Pol IV (III→IV), Pol IV to Pol IIIcore (IV→III), or co-localization (III+IV)"). In Table 2 we have inserted a division between the first six lines that record the exchange between Pol IIIcore and Pol IV, and the last two lines that record the exchange between Pol IIIcore and Pol II.